# Profound Non-Randomness in Dinucleotide Arrangements within Ultra-Conserved Non-Coding Elements and the Human Genome

**DOI:** 10.3390/biology12081125

**Published:** 2023-08-12

**Authors:** Larisa Fedorova, Emily R. Crossley, Oleh A. Mulyar, Shuhao Qiu, Ryan Freeman, Alexei Fedorov

**Affiliations:** 1CRI Genetics LLC, Santa Monica, CA 90404, USA; lvfedorova3@gmail.com (L.F.); amulyar@crigenetics.com (O.A.M.); rfreeman@crigenetics.com (R.F.); 2Program of Bioinformatics and Proteomics/Genomics, University of Toledo, Toledo, OH 43606, USA; emily.crossley@rockets.utoledo.edu; 3Department of Medicine, University of Toledo, Toledo, OH 43606, USA; shuhao.qiu@utoledo.edu

**Keywords:** bioinformatics, genomics, inhomogeneity, polymorphism, DNA structure, stacking

## Abstract

**Simple Summary:**

Our study aims to understand mysterious parts of our DNA, known as ultra-conserved non-coding elements. Found in the genomes of mammals and other vertebrates, these long (300 nucleotides, on average) DNA fragments have remained unchanged for hundreds of millions of years, despite numerous mutations happening within them. We do not yet know the function of these elements, but their extraordinary evolutionary stability could help us devise new approaches to preventing harmful mutations that lead to cancer and genetic disorders. We sought to discover why ultra-conserved non-coding elements are so resistant to change by studying their smaller building blocks, called dinucleotides, which play a crucial role in the shape and structure of DNA. Using a bioinformatics approach, we compared the differences in arrangement of dinucleotides in the ultra-conserved non-coding elements to the rest of the human genome. Our study revealed unique patterns within the ultra-conserved non-coding elements sections that distinguish them from the rest of our genetic code.

**Abstract:**

Long human ultra-conserved non-coding elements (UCNEs) do not have any sequence similarity to each other or other characteristics that make them unalterable during vertebrate evolution. We hypothesized that UCNEs have unique dinucleotide (DN) composition and arrangements compared to the rest of the genome. A total of 4272 human UCNE sequences were analyzed computationally and compared with the whole genomes of human, chicken, zebrafish, and fly. Statistical analysis was performed to assess the non-randomness in DN spacing arrangements within the entire human genome and within UCNEs. Significant non-randomness in DN spacing arrangements was observed in the entire human genome. Additionally, UCNEs exhibited distinct patterns in DN arrangements compared to the rest of the genome. Approximately 83% of all DN pairs within UCNEs showed significant (>10%) non-random genomic arrangements at short distances (2–6 nucleotides) relative to each other. At the extremes, non-randomness in DN spacing distances deviated up to 40% from expected values and were frequently associated with GpC, CpG, ApT, and GpG/CpC dinucleotides. The described peculiarities in DN arrangements have persisted for hundreds of millions of years in vertebrates. These distinctive patterns may suggest that UCNEs have specific DNA conformations.

## 1. Introduction

### 1.1. Ultra-Conserved Non-Coding Elements (UCNEs)

This paper is the next step in our long-term project on deciphering DNA biomarkers that make UCNEs unchangeable over hundreds of millions of years. Thousands of these long (300 bp, on average) UCNEs, found within the intergenic and intronic regions of mammals, birds, and other vertebrates, demonstrate identical orthologous sequences across species despite undergoing numerous mutations [1,2,3,4]. Interestingly, individual UCNEs show no sequence similarity to all other UCNE members within the same species. Recently, our team revealed that UCNEs are characterized by a higher abundance of GpC dinucleotides (DNs) and a depletion of GpG and CpC DNs [4]. Currently, this is the only reported distinction in nucleotide sequence between UCNEs and the rest of the genome. In this paper, we dive deeper into this phenomenon and focus on the unique peculiarities of global dinucleotide (DN) arrangement within UCNEs versus the entire human genome.

### 1.2. Levels of Genomic Sequence Non-Randomness

DNA sequences in mammalian genomes exhibit multiple levels of non-randomness in their nucleotide composition, as extensively discussed by Trifonov [5]. This non-randomness in nucleotide arrangement is highly complex and shares many similarities with the distribution of letters in the English language. Both nucleotide sequences and letters are forms of information character strings that demonstrate evident preferences or aversions for neighboring characters. For instance, in English, the letter “a” is one of the most frequently used characters, but it never occurs consecutively (“aa”) except in a few words (aardvark, etc.). On the other hand, the letter “u” strongly prefers to follow the letter “q”. The same phenomenon of preferences and aversions exists within genomic sequences. For example, in mammals, the “CpG” dinucleotide is very rare, while the “ApG” dinucleotide is one of the most common; nucleotide G has a preference to be behind A, forming “ApG”. The statistical measurement of dinucleotide non-randomness is known as “genomic signatures” [6] and is species-specific. These influences, or correlations, between neighboring nucleotides decrease significantly when they are separated by one character, and practically vanish when they are separated by two or more characters [7]. Now, let us consider grouping neighboring characters. In the English language, only a fraction of possible letter combinations forms a real word. For instance, the letters M, A, N, and Y can form the word MANY, but not MYAN, MNYA, etc. For DNA, the situation is the opposite: Any arrangement of ten nucleotides occurs in the human genome as 10-mer oligonucleotides, which, in total, presents 4^10^ combinations. The next level of non-randomness is represented by preferences of words to be associated in strings with each other. A particular word may affect the probabilities of other words within an entire paragraph. For instance, the presence of word “galactosidase” increases the chances of other biochemical terms appearing in the whole paragraph, while reducing the probabilities of many specific words from Jane Austen vocabulary in the same text. Similar vague influences exist among large groups of nucleotides in eukaryotic genomes, creating what are described as mid-range and long-range inhomogeneities in nucleotide compositions, reviewed in Fedorova and Fedorov [8]. These inhomogeneities create different DNA conformations (e.g., Z-DNA or H-DNA). Numerous and weak mid-range genomic signals are incredibly difficult for interpretation and implementation for practical goals. Thus, this area of Genomics rarely appeared in mainstream science. Yet, several publications over the last two decades have revealed different genomic sequence periodicities and non-randomness. For example, Bettecken and Trifonov 2009 [9] described periodicities of DN arrangements in multiple eukaryotic species and their association to nucleosome positioning. They discovered the most profound 10 bp periodicity in invertebrates, which is nearly absent in mammals. Additionally, peculiarities of DN distribution in the human genome were studied by Bastos et al., 2011 [10], while Cohen 2022 [11] investigated distinct signatures and codes within different genomic sequence locations of the human genome. Cohen’s study specifically focuses on the variation of DNs’ genomic profiles in coding, UTR, promoter, and non-coding sequences. Basu et al., 2021 [12] conducted experiments to estimate how the occurrence and spatial distribution of DNs impact DNA bendability. They demonstrated the importance of full turn helical period (10 bp) and half-helical period (5 bp) and how they result in globally curved or straight DNA molecules. Mathematical analysis of dinucleotide interactions and periodicities of DNA using physical approaches was performed by Valenzuela 2017 [13]. Different periodicities of genomic sequences in various bacteria were explored by Mrazek 2010 [14] and Kravatskaia et al., 2011 [15], while Frenkel et al., 2017 explored different periodicities in the large spectrum of eukaryotes and prokaryotes [16]. Atzinger and Lawrence 2020 [17] suggested that selection has likely been involved in the formation of ancient periodic DN spacing. The most extensively studied and profound nucleotide periodicity is the 10–11 bp periodicity, which is associated with a complete turn of a DNA helix (10.4 bases). This periodicity was first described by Trifonov and Sussman (1980) [18] and has been studied in multiple papers since then, including the work by Kumar et al. (2006) [19]. A public program known as *periodicDNA* in R-package was developed by Serizay and Ahringer (2021) [20] to enable in-depth statistical analysis of DNA periodicity. Finally, long-range periodicities, which are frequently associated with nucleosome positioning and GC-content, were described in several publications, including Traverse et al. (2010) [21] and Maqtaderi et al., 2021 [22].

In this paper we focus on DNs because they are the most important elements that influence DNA conformation and stability (see discussion section). Strong non-randomness in DN arrangements have been revealed.

## 2. Materials and Methods

### 2.1. Databases

We used our purified set of 4272 UCNE sequences described and available form Fedorova et al. [4]. This set was created from the human UCNE database (https://ccg.epfl.ch/UCNEbase/ (accessed on 6 August 2023)) [23].

Human genome sequence with masked repetitive elements (shown in small case letters) was downloaded from https://hgdownload.soe.ucsc.edu/downloads.html (accessed on 6 August 2023) UCSC genome browser as an assembly of the human genome (hg38, GRCh38 Genome Reference Consortium Human Reference 38, accession: GCA_000001405.15), accessed on 6 August 2023.

Chicken genome sequence with masked repetitive elements was downloaded from NCBI FTP server ftp://ftp.ncbi.nlm.nih.gov/genomes/ as RefSeq assembly accession: GCF_016700215.1 (version bGalGal1.pat.whiteleghornlayer.GRCg7w), accessed on 6 August 2023.

Zebra fish (*Danio rerio*) genome sequence with masked repetitive elements was downloaded from NCBI FTP server ftp://ftp.ncbi.nlm.nih.gov/genomes/ as RefSeq assembly accession: GCF_000002035.6 version GRCz11 (Genome Reference Consortium Zebrafish Build 11), accessed on 6 August 2023.

*Drosophila melanogaster* genome sequence with masked repetitive elements was downloaded from NCBI FTP server ftp://ftp.ncbi.nlm.nih.gov/genomes/ as an assembly: GCA_018904365.1 version ASM1890436v1, accessed on 6 August 2023.

Nucleotide and DN composition of our datasets were calculated using our previously published Perl program *NTcomposition.pl* [24]. Quasi-random subsets of nucleotide sequences with the same DN frequencies as within input table (for UCNE or WG) have been generated by our Perl program *SRI_generator.pl* published in [8] and explained in [25]. These programs are also available to run online via our web page: http://bpg.utoledo.edu/~jbechtel/gmri/, accessed on 6 August 2023. Each random subset has the same total nucleotide length as the UCNE dataset (1.4 million nucleotides).

All described nucleotide sequence datasets including 1000 random subsets are available from the Appendix A.

### 2.2. Programs for SNP Computational Processing

Distributions of spacing distances between DNs were computed by our Perl programs: *UCNE2_dint.pl*, *1000_dint.pl*, *master.pl*, *RUN_master.pl*, *RPD_calc.pl*, *RUN_ RPD_calc.pl*.

All Perl programs are available on our website (http://bpg.utoledo.edu/~afedorov/lab/UCNE2.html, accessed on 6 August 2023), in a package that includes an Instruction Manual (UCNE2instruction.docx). In addition, this package of programs and instructions is available in the Appendix A.

### 2.3. Statistics

Standard deviation (σ) and average values for DN spacing distances were calculated from 1000 subsets of WG, randWG, and randUCNE sequences. These statistics have been calculated using Perl module Statistics::Basic (Version 1.6611), (Miller, P. 2014, Retrieved from https://metacpan.org/dist/Statistics-Basic/view/lib/Statistics/Basic.pod, accessed on 6 August 2023) inside our Perl program *1000_dint.pl.* The σ for 1000 random subsets (randUCNE, randWG) is solely due to the limited size of each subset (1.4 million nucleotides). In contrast, σ for 1000 WG subsets is partially due to the limited size of each subset (which has the same size of 1.4 million nucleotides), plus variations in DN frequencies distributions and GC-content in each WG subset. Comparison of σ of WG versus randWG demonstrated that σ_WG_ is only slightly larger than σ_randWG_ (by no more than 20%). Hence, the most impact on σ comes not from the fluctuations of GC-composition of individual WG sequences but due to the limited size of the sample. Therefore, we assumed that σ_UCNE_ should be the same as σ_WG_, because they have the same size of 1.4 million nucleotides and both subsets have variations in nucleotide compositions of individual sequences. This conjecture has been tested via the comparison of UCNE DN spacing distributions on complementary strands (such as TT-GA versus TC-AA DN spacing distributions, as demonstrated in the results section). Examinations of complementary pairs demonstrated that our σ_UCNE_ evaluations are correct, and the fluctuations are in 68% of the time within 1 σ. These 3xσ_UCNE_ are plotted in the figures in the Results section.

## 3. Results

Our algorithm for counting spacing distances between DNs is explained in Figure 1. Figure 1A presents a spacing scheme for the same DNs (in this example, GpC-GpC distances), while Figure 1B shows the algorithm for distances between two different DNs (GpC–TpC). We considered only the DNs closest in proximity to each other. Due to this condition, we ignored the second and the fourth TpC DNs in Figure 1B because the first and third TpC DNs are located between them and the nearest GpC on the left side, respectively. For the same reason, we also ignored the first and fourth GpC DNs because the second and fifth GpC DNs are closer to neighboring TpC on the right side, respectively. In other words, for GpC -> TpC distances, we considered all DNA fragments that contain GpC at the 5′-terminus and TpC at 3′-terminus and exclude GpC and TpC DNs inside them.

For the evaluation of non-randomness in spacing distances between DNs, we chose a computational approach over advanced statistics. Under this methodology, we computationally generated quasi-random sets of nucleotide sequences that have the same DN frequencies as a natural UCNE dataset of sequences (so-called randUCNE sets) or whole genome (WG) unique sequences (randWG sets). To decrease standard error, 1000 independent quasi-random subsets of these sequences have been generated and processed. Each random set has the same total sequence length as the UCNE dataset. Since the unique sequences of the whole human genome are one thousand times larger than UCNE dataset, we also generated 1000 randomly chosen subsets of whole genome sequences (WG datasets), each of the same size as the UCNE dataset. The computation of 1000 datasets allows one to (1) calculate the standard deviation (σ) for spacing distances and (2) average the results and minimized standard error of the mean (SE) by SE = $\sigma /\sqrt{1000}$. The distribution of DN spacing distances for four types of data (UCNE, randUCNE, WG, and randWG) has been calculated by our Perl programs, and the results for a GpC-TpC DN pair is exemplified in Table 1. The entire set of these results for all 256 possible DN pairs are in Appendix A. A statistically significant difference (over 3σ, *p*-value 0.003) in the DN spacing distances has been observed between the natural sequences (UCNE or WG) and the corresponding random datasets (randUCNE or randWG) for a majority (≥83%) of DN pairs (see Table 2). Additionally, there is a significant difference between UCNE and WG. For example, in Table 1, for distance L = 4 nt, we observed a 7% difference (peak) between UCNE and randUCNE, which is over 4 times the standard deviation (*p*-value < 0.001). For L = 3 nt, there is a negative −19% difference (dip) between UCNE and randUCNE, which is over 12 times the standard deviation. These differences (RPD values) were calculated using Formula (1). Furthermore, for L = 4 nt, there is an 18% difference between UCNE and WG according to Formula (1), which is partially due to the nucleotide composition difference between these two samples. Therefore, we calculated the RPD between UCNE and WG using Formula (2), which results in RPD_4_ = 10%. Figure 2 illustrates prominent differences for DN spacing distances, with Figure 2B representing the results from Table 1. In a majority of cases from Figure 2, the statistically significance is over 5σ, *p*-value 3 × 10^−7^.

Figure 2A shows a significant difference in spacing between neighboring GpC DNs among UCNE, WG, and random sequences. All four curves have a minimum at L = 2 nt due to the formation of a rare CpG DN in the tetramer “GCGC” formed by two adjacent GpC DNs. In this figure, quasi-random datasets exhibit monotonously decreasing curves from L = 3 nt and onwards. In contrast, the WG curve has a peak at L = 5, which ends at L = 8. In contrast, the UCNE curve has a dip at L = 6. Both WG and UCNE exhibit a peak at L = 3 nts, for which the value is 30% above the random curves (RPD_L=3_ > 30%). Figure 2B–D demonstrate a peak in UCNE DN spacing distance at L = 4 nt, which is absent in WG, randUCNE, and randWG curves. Figure 2B,C represent complementary strands (TA-GC vs. GC-TA), demonstrating identical curves within statistical standard deviation and standard error intervals. Figure 2C,D have similar peaks for the L = 4 UCNE curve, representing similar DN pairs (GpC -> TpA vs. GpC -> TpT) that differ only in the last nucleotide (A vs. T). This similarity suggests that there are some hidden biological sense and rules for non-randomness in DN spacing. Figure 2E–I illustrate a variety of patterns of non-randomness in DN spacing distribution observed in UCNE and WG curves, with the most prominent differences observed in the spacing length interval from L = 2 to L = 6.

The non-randomness in DN spacing within UCNE compared to randUCNE has been measured as RPD using Formula (1). The same formula was used to evaluate non-randomness in DN spacing between WG and randWG. However, because UCNE and WG have different DN frequencies and GC-content (38.6% and 41.7%, respectively), it is inappropriate to measure the difference in DN spacing between them using Formula (1). Thus, the non-randomness between UCNE and the WG has been calculated using Formula (2), which considers the dissimilarity in DN frequencies between these two samples. RPDs were calculated using Formulas (1) and (2) for each specific DN spacing distance (L), such as RPD_5_ for L = 5 nucleotides and RPD_7_ for L = 7 nucleotides.
(1)$$RPD_{L}=\frac{N_{L,\;WG}-N_{L,\;rand\;WG}}{\left(N_{L,\;\;WG}+N_{L,\;randWG}\right)÷2}\times 100\%$$
(2)$$RPD_{L}=\frac{\left(N_{L,\;WG}-N_{L,UCNE}\right)-\left(N_{L,\;randWG}-N_{L,\;randUCNE}\right)\;}{\left(N_{L,\;WG}+N_{L,\;UCNE}\right)÷2}\times 100\%$$
where *N_L_*_,*WG*_ is the number of observations for the *WG* dataset for a particular spacing distance L.

When RPD exceeds 10%, the difference is statistically significant with at least 3 times the standard deviation certainty (*p*-value 0.003). For many DN pairs with this threshold, statistical significance is over 5σ (*p*-value 3 × 10^−7^). Table 2 presents the fraction of all possible DN pairs where RPD exceeds 10%,20%, 30%, and 40% for at least one spacing distance *L* in the range of 2–12 nts. Table 2 demonstrates that non-randomness in DN spacing arrangements exists for ≥83% of DN pairs with RPD > 10%. Additionally, in over one-third of the possible DN pairs, the RPD reaches 20% and, occasionally, even exceeds 40% of RPD values. This suggests that the non-randomness in DN spacing is comparable to the non-randomness of genomic signatures for adjacent nucleotides, as introduced by Karlin and Burge [6]. Moreover, the strongest effect of DN spacing is not limited to adjacent DN pairs (*L* = 2 nt) but frequently occurs at other distances within the range of 2–7 nucleotides. For example, in Figure 2B–D, UCNE exhibits RPD_L=4_ peaks of approximately 20% over WG, while Figure 2E shows a UCNE dip for RPD_4_. To account for the effect of the distance *L* on the non-randomness of DN spacing, we calculated at which *L* value a maximum (peak) and minimum (dip) are observed for each DN pair, as described in the Materials section. These calculations are shown in Figure 3. On average, peaks or dips in 40% of the cases occur at *L* = 2, in 29% of the cases at *L* = 3; in 16% at *L* = 4; in 10% at *L* = 5; and 5.7% at *L* = 6.

The list and number of occurrences of DN from pairs with highest (>30%) RPD values are listed in the Table 3. The most frequent DNs for UCNE vs. randUCNE are GpC, ApC, and GpT. The most frequent for WG vs. randWG are CpG and GpC, and for WG vs. UCNE the most frequent DNs are CpG and ApT. The rarest DNs to form significant non-randomness for all three comparisons are GpA and TpC. Interestingly, CpC and GpG DNs present frequent significant non-randomness for WG vs. randWG, but not for UCNE vs. randUCNE.

Finally, we calculated the non-randomness of DN spacing for WG of other species (chicken, zebra fish, and fruit fly) and compared them with human. The results are presented in the Appendix A, while one example for the GpC-GpC DN pair is illustrated in the Figure 4. This figure demonstrates that the DN spacing non-randomness is practically identical between human and chicken. In zebrafish, we observed the overall similarity to the human-chicken non-randomness, yet with noticeable fluctuations. However, for the Drosophila genome the distribution pattern significantly changes. Specifically, a prominent dip at *L* = 4 is absent, while species-specific small peaks at *L* = 6 and *L* = 9 are detected in fruit fly (see Figure 4).

## 4. Discussion

### 4.1. DN Stacking Energy

The stacking interaction of adjacent nucleotides located on the same strand is the strongest type of non-covalent bond that creates double-stranded helical DNA conformation [26]. Notably, the stacking interaction is formed by the assemblage of DNs in the DNA sequence. The strength of stacking varies by two magnitudes for different DNs [27,28,29,30]. The strongest stacking force is formed by the GpC DN, while some of the weakest are formed by GG and CC DNs [31,32,33]. However, there are ongoing controversies in measuring stacking forces, and different experimental methods may produce diverse results [26]. The importance of DN composition for DNA structures can be demonstrated by the following thought experiment. Imagine a 1000-nucleotide sequence with equal numbers of the four bases (A, G, T, and C), randomly distributed. This sequence would likely adopt the B-form DNA conformation under physiological conditions [34]. Now, arrange the same bases in a specific order, having them alternating purines (R) and pyrimidines (Y), for example, “RYRYRY…”. This second sequence presumably should form Z-DNA with a drastically different conformation, including an opposite helical rotation [35,36]. Lastly, let us create another arrangement of the same group of nucleotides, with all purines on one side of the strand and pyrimidines on the other side. This third sequence will likely form DNA triplexes or H-DNA conformations [37]. All three imaginary DNA molecules have exactly the same nucleotide composition but exhibit radically different DN compositions. Thus, the composition and arrangement of DNs is much more critical for various DNA shapes than overall nucleotide composition.

Last year, our team demonstrated that UCNEs have unusual DN frequencies and contain higher amounts of GpC DNs [4]. In this paper, we hypothesized that unique DN arrangements may exist within UCNEs, contributing to their evolutionary conservation and investigated this conjecture.

### 4.2. Non-Randomness in DN Spacing Arrangements

One of the strongest and most studied nucleotide periodicities in the genomes of vertebrates is the 10 bp periodicity of “weak” DNs formed by A and/or T bases [5,38]. This periodicity aligns with a complete turn (10.5 bases) of relaxed B-DNA helix [17] creating a specific curvature of DNA that helps in the formation of nucleosomes on chromosomal DNA. In mammals, the amplitude of this 10-bp periodicity does not exceed ±4% [17,20]. Our analysis of spacing distances between all possible DN pairs revealed significant non-randomness in their arrangements, which often exceeds 10% of RPD and even reaches 40% in extreme cases (Table 2). Thus, our results present the strongest form of non-randomness in DN arrangements among vertebrates. Recently, Basu et al. demonstrated that DN pairs at distances corresponding to half-turns of a DNA helix (5 base pairs) are crucial for DNA curvatures [12]. In our calculations, we frequently observed peaks and dips for arrangements of DN pairs at the distance of *L* = 4 nt (see Figure 2B–D for example). According to our DN spacing counting scheme (Figure 1), the distance *L* = 4 approximately corresponds to a half turn of a DNA helix (10.5 bases). This suggests a potential association between non-randomness in DN spacing and local DNA curvatures and conformations within the range of *L* = 3 to *L* = 5.

In the genomic studies of spacing distances between oligonucleotides, the presence of DNA repetitive elements often produce prominent peaks but not dips [39]. Therefore, in our WG nucleotide datasets, we considered only unique DNA fragments selected using rigorous *RepeatMasker* computation. However, short, fuzzy, simple repeats may still be present, as escaped remnants inside masked DNA datasets, and the possible noise from short fuzzy simple repeats may lead to potential overestimation of DN spacing peaks at very short distances (*L* = 2 or *L* = 3) (Figure 3). Despite this, the presence of dips alongside peaks in the DN spacing distributions indicates that the observed non-randomness cannot be attributed solely to some hypothetical, ancient, repetitive DNA, which remained undetected in our datasets. Therefore, the observed DN spacing non-randomness represents a novel phenomenon that may be caused by mutational biases, functional selection, or other factors, requiring further investigation. The discovered spacing non-randomness forms weak yet numerous signals from over two hundred types of DN pairs. To fully understand their effects and predict their potential for classifying UCNEs, a specialized bioinformatics investigation is necessary. This presents an opportunity for a specific research project.

Mammals and other multicellular organisms have a complex, hierarchical, 3D genome organization, characterized by remarkable global chromosomal compartments divided into specific segments and further into smaller blocks [40,41,42]. Various genomic regions have different global spatial structures, epigenomic patterns, coding properties, and spectra of associated proteins. At the DNA sequence level, such hierarchical structures are maintained through different nucleotide compositions, DNA folding, and diverse distributions of cis-regulatory elements [41,43]. DN local non-randomness and specific distributions likely play important roles in chromosomal assembly and functioning. Thus, further investigations of DN arrangements are crucial for understanding genome structure and evolution.

## 5. Conclusions

Last year our team found unusual accumulation of GpC DNs within UCNEs. In this paper we demonstrated that GpC DNs also have distinct patterns of spacing arrangements inside UCNEs in contrast to the rest of genomic sequences. Moreover, we found that 83% of all DN pairs have significant (>10%) non-random genomic arrangements at short distances (2–6 nucleotides) relative to each other. Similar patterns of non-randomness in DN arrangement within the human genome were also observed in birds and other vertebrates.

Human UCNEs have significant (>10%) distinction in DN arrangement compared to the whole genome for 90% of all DN pairs. Non-randomness in the arrangement of DN pairs inside UCNEs represents hundreds of weak signals that likely contribute to the complex folding organization of chromosomes, creating structural differences between UCNEs and the rest of the genome. The strongest non-randomness has been observed for GpC, CpG, ApT, and GpG/CpC DNs. The described non-randomness for numerous DN pairs spacing arrangements between UCNE and the WG allows for using these data for the prediction of UCNEs using AI algorithms.

## Figures and Tables

**Figure 1 biology-12-01125-f001:**
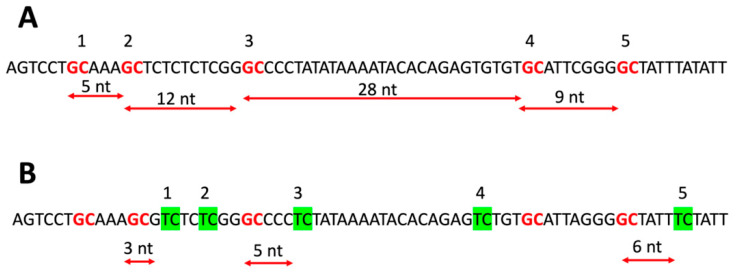
Algorithm for calculation of distances between DNs. (**A**) delineates the distances for the same type of DN that forms GpC -> GpC pairs. (**B**) delineates on the same sequence the distances for two different DN types that form the closest GpC -> TpC pairs. DNs are shown in red (GpC) and green (TpC) colors and have numbers above them. The distances are measured in nucleotides (nts). Note that examined DN pairs have direction on DNA sequence. For GpC -> TpC pairs, the GpC should always be at the 5′-end and TpC on the 3′-end (**B**).

**Figure 2 biology-12-01125-f002:**
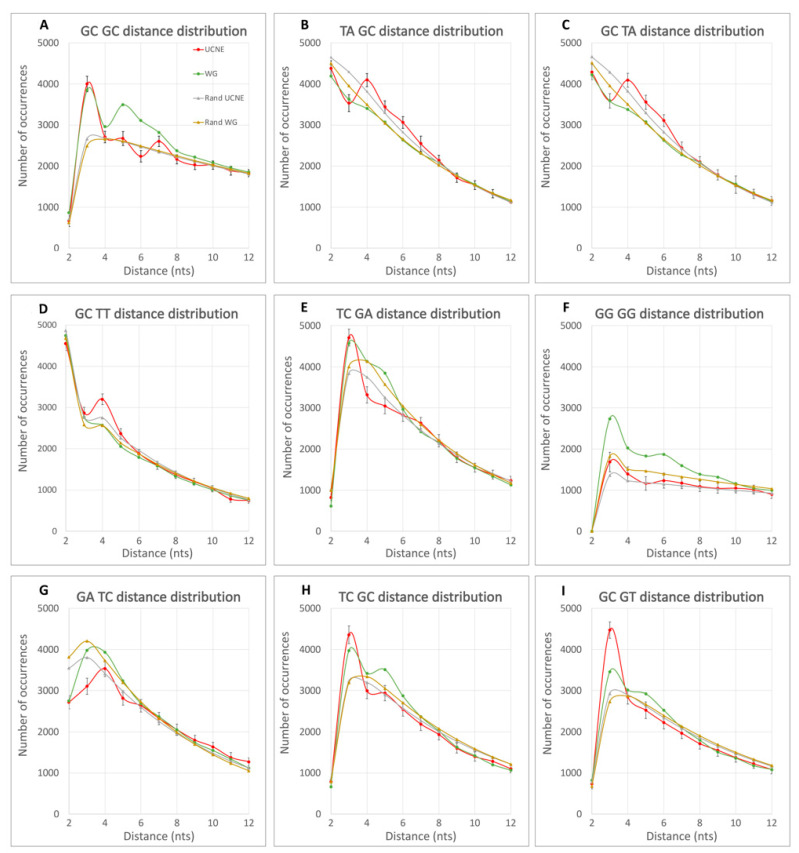
Distribution of the DN spacing distances for UCNE (red), WG (green), randUCNE (yellow), and randWG (gray). The 99.7% confidence intervals (±3σ) are demonstrated for UCNE datasets as vertical bars. Statistical errors for averaged WG, randWG, and randUCNE are 30 times less than standard deviation for UCNE, and are invisible in these graphs.

**Figure 3 biology-12-01125-f003:**
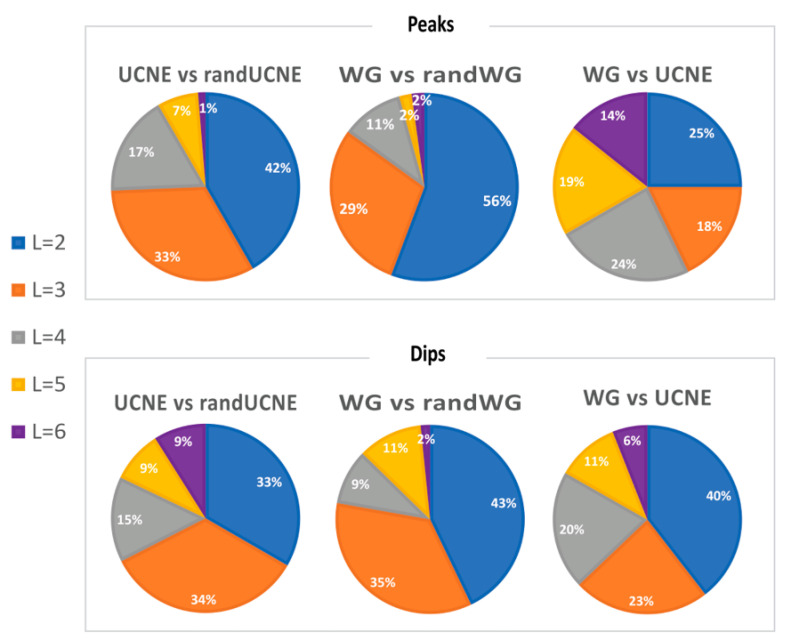
Fractions of DN pairs for which most prominent peaks or dips were observed for a specific spacing distance L in the range from 2 to 6 nucleotides.

**Figure 4 biology-12-01125-f004:**
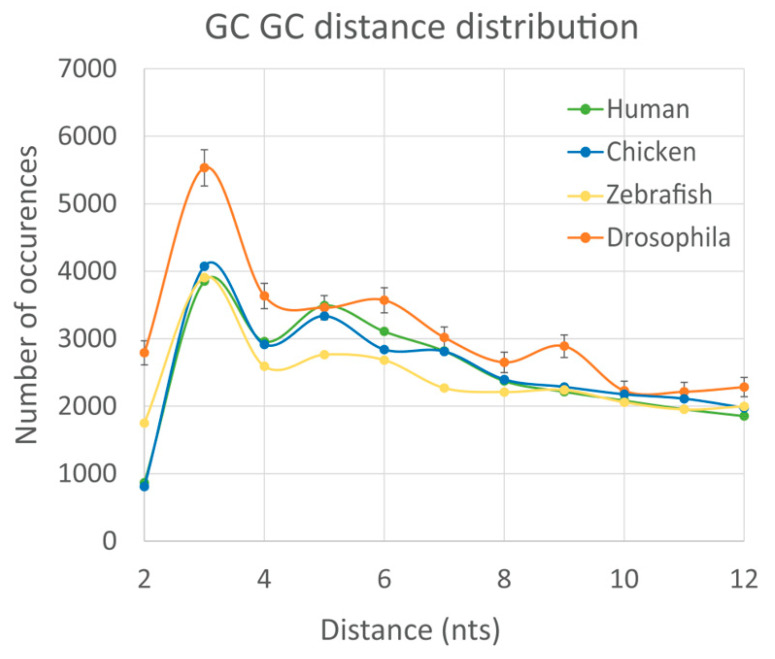
Distribution of the DN spacing distances for human WG (green), chicken WG (blue), zebrafish WG (yellow), and *Drosophila melanogaster* WG (orange). The results are averaged from the 1000 random WG subsets of these species, so the standard error is 31 times smaller than standard deviation.

**Table 1 biology-12-01125-t001:** This table represents the distribution of spacing distances between TpA -> GpC DNs. Column 1 displays the distance between TpA and GpC measured in nucleotides (nts). Column 2 shows the total number of observed DNA fragments for that specific distance (nts) between TpA and GpC DNs in the entire UCNE dataset. Columns 3–5 show the average number of observed DNA fragments from 1000 independent datasets (randUCE, WG, and randWG respectively). Columns 6–8 present standard deviation (SD) calculated from 1000 independent datasets.

Distance	UCNE	WG, Average	randUCNE, Average	randWG, Average	WG	randUCNE	randWG
(nts)	#Observations	#Observations	#Observations	#Observations	SD	SD	SD
1	0	0	0	0	0	0	0
2	4380	4186	4650	4499	62.3	62	62.9
3	3531	3625	4292	3952	62.7	60.9	61.1
4	4096	3401	3818	3499	68.4	59.6	55.6
5	3440	3066	3295	3044	53.3	55.7	55.3
6	3064	2634	2826	2656	49.2	52.2	49.7
7	2545	2305	2422	2319	47.9	44.9	46.8
8	2138	2082	2074	2021	61.7	44.3	43.9
9	1716	1779	1778	1760	41.9	40.3	41.2
10	1537	1555	1522	1537	37.9	37.9	37.1
11	1325	1333	1304	1341	36.2	37.5	35.2
12	1115	1138	1117	1171	33.5	34.5	33.9
…							
50	6	12	3	7	3.4	1.8	2.6

**Table 2 biology-12-01125-t002:** Fraction of DN pairs that exhibit significant non-randomness in spacing distances between these DNs at least for one distance L in the range from 2 to 10 nucleotides. Three types of comparisons (UCNE vs. randUCNE; WG vs. randWG; and WG vs. UCNE) are presented. Non-randomness is measured in absolute values of RPD. When (N_UCNE_ > N_randUCNE)_ the RPD value is positive (maximum) and is visible as peaks on the graphs (e.g., Figure 2C). When (N_UCNE_ < N_randUCNE)_ the RPD value is negative (minimum) it is visible as dips on the graphs. Peaks and dips are counted separately (columns 2–4 and 5–7, respectively) and together (columns 8–10). Fractions of DN pairs producing non-randomness have been counted for four different thresholds for absolute values of RPD: >10%, >20%, >30%, and >40%.

abs(RPD) Threshold	MAX (Peaks)			MIN (Dips)			TOTAL (Peaks + Dips)		
	UCNE-Rand	WG-Rand	WG-UCNE	UCNE-Rand	WG-Rand	WG-UCNE	UCNE-Rand	WG-Rand	WG-UCNE
10%	69%	62%	56%	73%	54%	68%	90%	83%	84%
20%	27	34	7	16	14	19	40	44	25
30%	9	13	0	2	3	8	11	17	8
40%	4	7	0	0.4	0.4	0.8	4	7	1

**Table 3 biology-12-01125-t003:** Number of occurrences of dinucleotides with RPD values more than 30% for three comparisons: UCNE vs. randUCNE, WG vs. randWG, and WG vs. UCNE. The total in the last row represents the number of DN pairs that have an RPD value > 30% for the comparison.

DN	UCNE vs. randUCNE	WG vs. randWG	WG vs. UCNE
AA	3	2	4
AG	4	3	0
AC	6	3	1
AT	2	7	6
GA	0	3	0
GG	3	8	3
GC	10	9	2
GT	6	3	1
CA	3	4	1
CG	5	11	7
CC	1	8	4
CT	4	3	0
TA	2	5	5
TG	3	4	0
TC	0	3	0
TT	4	3	3
TOTAL	29	43	21

## Data Availability

All Appendix A and Perl programs are available on our website (http://bpg.utoledo.edu/~afedorov/lab/UCNE2.html, accessed on 30 June 2023) in a package that includes an Instruction Manual (UCNE2instruction.docx) and Protocols (UCNE2protocols.docx).

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
