# Peer review of "Profound Non-Randomness in Dinucleotide Arrangements within Ultra-Conserved Non-Coding Elements and the Human Genome"

_biology, 2023, doi:10.3390/biology12081125_

Round 1

Author Response

Our reply is attached

Reviewer 2 Report

Considering next neighbour distances of DN as field of investigation is an interesting and good approach.

line 32

"significant (>10%) non-random genomic arrangements"

you specify in line 160 how you get 10% and also refer later in Table 2 to abs(RPD) threshold

in the abstract it is not clear what you mean by this ">10%" and how you obtain it

Line 50

"and the rest of the genome"

according to reference 4 this has been based on a comparison of UCNEs to whole rest of the genome but not on individual subsets as CpG islands

is there any information available on such tiny subsets of the genome in which these features might also be present but would not become visible if analysed with all "the rest of the genome"?

line 144

you reduce the standard deviation (sigma) by sqrt(1000) and you get SE

table 1

what is "av" in 3-5?

would a switch in 6-8 of first and second line be better

(WG in first line and SD in second?)

line 175

WH?

line 181

"only in the forth"

ambigous, as it is the forth letter of DN GpC plus DN TpA, but it is L=4 and so it would be GCNNTA vs. GCNNTT and so would be sixth

line 192

you describe

relative percentage difference (RPD)

but you already use it in line 156

line 218 and 219

pick?

Table 3 and Figure 3

RAND is always respective RAND of set given?

so UCNE vs. RAND is UCNE vs. RANDUCNE?

Figure 4

order in legend may be changed based on taxonomy or other order

line 270

reference for the B-DNA?

line 292

often exceeds 10% and even reaches 40% in ex-treme cases

may be you add some info based on RPD

line 298

distance L=4 corresponds to a half turn

it puts your DN in the center of a half turn after completing 5bp (one nt before and one after the half)

and so you come up to L=3 to L=5 relating your DN to a half turn

(maybe just a few sentences for explanation)

line 302 pick

line 315

you may use AI if you want, it is not necessary

on what will you train?

line 317

why now just mammals?

you now try to focus on the relevance of DNs...you may add just a sentence before this paragraph to smoothen it for the reader and explain why you do focus

line 388

is less than 20% bigger of

than?

comes not

does not...

line 392

may you explain with a few sentences why this is sufficient for yor testing?

and why now 100?

very few errors

spelling

article

Author Response

Our reply is attached

Reviewer 3 Report

The paper titled 'Profound Non-Randomness in Dinucleotide Arrangements within Ultra-conserved Non-Coding Elements and the Human Genome' by Fedorova and co-authors presents an intriguing exploration of a complex genomics issue. The authors discovered a unique pattern in non-coding sections of Ultra-conserved Non-Coding Elements (UCNE). Their approach is well-executed, but one limitation is the lack of direct connection with experimental data.

In the discussion, the authors suggest that : ".DN local non-randomness and specific distributions likely play important oles in chromosomal assembly and functioning.". To further validate their findings, it might be worthwhile for the authors to explore publicly available data, such as ChipSeq or ATAC-seq, which could complement their analysis.

While UCNEs are primarily discussed in vertebrates, the inclusion of arthropods in the study introduces an interesting perspective. The slight variation observed in Drosophila could be attributed to the differences in GC content, which is known to vary among arthropods. To bolster this finding, it could be beneficial to compare the results with other well-known arthropod model organisms, such as Tribolium castaneum and Apis mellifera. Otherwise the necessity of including Drosophila in the paper is not quite clear. 

Author Response

Our reply is attached
